

# A comparison of chain-of-thought reasoning strategies across datasets and models

Konstantin Hebenstreit[1,*], Robert Praas[2,*], Louis P. Kiesewetter[3] and Matthias Samwald[1]

[1] Medical University of Vienna, Center for Medical Data Science, Institute of Artificial Intelligence, Vienna, Austria
[2] KTH Royal Institute of Technology, Stockholm, Sweden
[3] Humboldt University, Berlin, Germany
[*] These authors contributed equally to this work.

## ABSTRACT

Emergent chain-of-thought (CoT) reasoning capabilities promise to improve the performance and explainability of large language models (LLMs). However, uncertainties remain about how reasoning strategies formulated for previous model generations generalize to new model generations and different datasets. In this small-scale study, we compare different reasoning strategies induced by zero-shot prompting across six recently released LLMs (davinci-002, davinci-003, GPT-3.5-turbo, GPT-4, Flan-T5-xxl and Cohere command-xlarge). We test them on six question-answering datasets that require real-world knowledge application and logical verbal reasoning, including datasets from scientific and medical domains. Our findings demonstrate that while some variations in effectiveness occur, gains from CoT reasoning strategies remain robust across different models and datasets. GPT-4 benefits the most from current state-of-the-art reasoning strategies and performs best by applying a prompt previously discovered through automated discovery.

## INTRODUCTION

Emergent chain-of-thought (CoT) reasoning capabilities in large language models (LLMs) promise to improve both predictive performance and explainability of models when applied to complex tasks (*Wei et al., 2021*). While good performance can be achieved by few-shot in-context prompting with exemplars suitable to a specific task, zero-shot prompting setups do not require task-dependent selection of exemplars (*Kojima et al., 2022*). The recent success of models optimized for dialog, such as GPT-3.5, further increases the expectation that models reach robust performance with ad-hoc reasoning strategies and are less influenced by minor variations. This study empirically investigates how previously discovered zero-shot CoT prompting styles generalize to new model generations and datasets and how they compare to newly developed reasoning strategies. We conduct our evaluations on six question-answering datasets of varying levels of complexity, including

Corresponding author
Matthias Samwald,
matthias.samwald@meduniwien.ac.at

**Table 1  Dataset descriptions.**

| Dataset | Description |
|---|---|
| CommonsenseQA | General domain crowd-sourced questions with high semantic complexity that command the use of prior knowledge. |
| StrategyQA | General domain crowd-sourced questions that require implicit reasoning and multi-step answer strategies. Yes/No answers. |
| WorldTree v2 | Elementary science questions for 3rd to 5th-grade level, combining domain-specific and world knowledge. |
| OpenBookQA | Scientific and broad common knowledge questions that require multi-step reasoning and rich text comprehension. |
| MedQA | Questions from medical board exams. We used only examples from the US (USMLE subset). |
| MedMCQA | Real-world medical entrance exam questions. |

scientific and medical domains. Portions of this text were previously published as part of a preprint (https://arxiv.org/abs/2305.02897).

# METHODS

## Datasets

For our study, we used the ThoughtSource framework (*Ott et al., 2023*), which provides a comprehensive meta-dataset and software library designed for streamlined generation, evaluation, and annotation of chain-of-thought (CoT) reasoning. We focused on real-world datasets that combined knowledge application with logical, verbal reasoning. We covered a sizable range of topics and complexity levels by selecting subsamples of six question-answering datasets spanning common-sense (*Talmor et al., 2019*; *Geva et al., 2021*), scientific (*Xie et al., 2020*; *Mihaylov et al., 2018*), and medical domains (*Jin et al., 2021*; *Pal, Umapathi & Sankarasubbu, 2022*) (Table 1). These datasets were multiple-choice, consisting of two to five answer options with a single correct response.

We used a template to structure the input handed to the model. Each of our chain-of-thought prompts is placed before the question, which we call "instruction," or after the question, which we call "cot trigger". Table 2 shows the exact template. An extra new line was added after the instruction or before the cot-trigger to separate the chain-of-thought prompt from the question.

## Prompts

We assembled a set of ten zero-shot reasoning strategies (Table 3) consisting of one baseline, two pre-existing, and seven novel designs:

1. Direct prompting: No specific trigger, serving as a baseline for comparison.
2. Kojima: A well-established CoT prompt, "Let's think step by step." (*Kojima et al., 2022*)
3. Zhou: An enhanced version created through automated prompt engineering, "Let's work this out in a step by step way to be sure we have the right answer." (*Zhou et al., 2023b*)

**Table 2 Comparison of prompt locations.** Each prompt is used either at the end (cot-trigger) or at the beginning (instruction), with its type detailed in Table 3 as parentheses. Example from CommonsenseQA.

| Chain-of-Thought prompt as Trigger | Chain-of-Thought prompt as Instruction |
|---|---|
| Where on a river can you hold a cup upright to catch water on a sunny day?<br>A) waterfall<br>B) bridge<br>C) valley<br>D) pebble<br>E) mountain<br><br>**Answer: Let's work this out in a step by step way to be sure we have the right answer.** | **Instruction: Let's work this out in a step by step way to be sure we have the right answer.**<br><br>Where on a river can you hold a cup upright to catch water on a sunny day?<br>A) waterfall<br>B) bridge<br>C) valley<br>D) pebble<br>E) mountain |

**Table 3 Prompt descriptions.** Prompt names with temples in brackets (see Table 2 for details) and their corresponding text.

| Prompt name | Text |
|---|---|
| *Direct* | Direct prompting. No specific prompt is used. Just the question and answer choices are the input to the model. |
| *Kojima*<br>(cot-trigger) | "Answer: Let's think step by step." |
| *Zhou*<br>(cot-trigger) | "Answer: Let's work this out in a step by step way to be sure we have the right answer." |
| *Zhou-instr.*<br>(instruction) | "Let's work this out in a step by step way to be sure we have the right answer." |
| *Plan*<br>(instruction) | "First think step by step - describe your plan for how to get to the right answer, written out in great detail. Then answer the question." |
| *Articulate*<br>(instruction) | "Carefully read the question & work this out in a step by step way to be sure you have the right answer. Be certain to spell out your thoughts & reasoning so anyone can verify them. Spell out everything in painstaking detail & don't skip any steps!" |
| *Rephrase*<br>(instruction) | "Instruction: First let's rephrase the question to be sure we understood it correctly. Second, let's work this out step by step by spelling out our thoughts & reasoning so anyone can verify them. Third, make sure we have the right answer." |
| *Elaborate*<br>(instruction) | "Answer the following question through careful, concise step-by-step reasoning. First, complement the question with helpful knowledge and important additional facts. Second, generate sub-questions that are required to answer the original question, answer them until you can answer the original question." |
| *Converse*<br>(instruction) | "Create a dialog between a professor and a student. The student asks sub-questions to the question. The professor works them out in a step by step way and makes sure that the student understood how they got to the right answer." |
| *Self-critique*<br>(instruction) | "Answer the question, then critique the answer. Based on the critique, reconsider the other answer options and give a single final answer." |

4. Seven original reasoning strategies we designed, inspired by various public resources (*OpenAI, 2023a*; *Schulhoff, 2022*), and refined through iterative adaptation based on analyzing outputs. One of these strategies employed a self-critique strategy, requiring the model to provide an initial answer, critique it, and then propose a revised response (*Madaan et al., 2023*; *Saunders et al., 2022*).

## Models

We included six instruction-tuned models based on their reported capabilities in CoT reasoning: davinci-002 (*Brown et al., 2020*), davinci-003 (*Ouyang et al., 2022*), GPT-3.5-turbo (*OpenAI, 2022*), and GPT-4 (*OpenAI, 2023b*) from OpenAI, Flan-T5-xxl from Google (*Chung et al., 2022*), and command-xlarge-nightly from Cohere (*Cohere.ai, 2023*). Between February and April 2023, we conducted 11,880 experiments, with the model temperature set at 0 for maximal determinism. We limited the output to 512 tokens to allow for thorough reasoning while preventing the occasional issue of infinite sequence repetition observed in smaller models. We used the LangChain framework (*Chase, 2022*) to access several APIs. Usage costs included: $190 for OpenAI's models through their API, Cohere's model for free *via* their API, and $30 for Flan-T5-xxl through a Hugging Face Inference Endpoint.

## Evaluation

We selected Krippendorff's alpha as our evaluation metric (*Krippendorff, 2011*). It allows for combining results from sub-datasets with different numbers of answer choices by correcting for their corresponding base probability rates. Krippendorff's alpha measured inter-rater reliability on a scale from zero (random chance) to one (complete agreement) and was used to compare model predictions to gold standard answers (*Castro, 2017*). We performed a power analysis using the formula below to determine an appropriate sample size.

$$T\left(P_c, \alpha_{\min}, p\right) = 2z_p^2 \left( \frac{(1+\alpha_{\min})(3-\alpha_{\min})}{4(1-\alpha_{\min})P_c(1-P_c)} - \alpha_{\min} \right)$$

where:

1. $P_c$ the probability of value $c$
2. $\alpha_{\min}$ the smallest $\alpha$ for coding to be accepted as reliable
3. $p$ level of significance
4. $z_p$ the standardized $z$-statistics at $p$.

We performed a power analysis with a significance level set at 0.05, a medium Krippendorff's alpha value of 0.8, and a base correct probability of 0.2, considering the maximum of five answer options in our sub-datasets. The analysis yielded a required sample size of 164 items, which we increased to 198 items, divided into six sub-datasets of 33 each. We used bootstrapping (r = 1,000) to compute means and confidence intervals for the generated results. To guarantee accurate Krippendorff scores, which depend on the number of options, we bootstrapped each sub-dataset individually when needed and calculated confidence intervals by pooling standard deviations.

**Table 4** **Performance of prompts.** Krippendorff's alpha ($\alpha$) performance of prompts averaged over datasets. Average taken solely for GPT-4 and over all six models, best results in bold. N total = 11,880.

| Prompt | GPT-4 $\alpha$ (CI) n per prompt = 198 | Model avg. $\alpha$ (CI) n per prompt = 1188 |
|---|---|---|
| Zhou | **.83** (.77, .90) | .53 (.50, .57) |
| Kojima | .80 (.73, .87) | .51 (.47, .55) |
| Zhou-instr. | .79 (.72, .86) | .50 (.46, .54) |
| Articulate | .79 (.71, .86) | .52 (.48, .56) |
| Rephrase | .78 (.71, .85) | **.54** (.51, .58) |
| Plan | .77 (.71, .84) | .50 (.46, .54) |
| Elaborate | .77 (.70, .84) | .51 (.47, .55) |
| Self-critique | .76 (.69, .84) | .49 (.45, .53) |
| Converse | .74 (.66, .81) | .47 (.43, .51) |
| Direct | .71 (.64, .79) | .49 (.45, .52) |

**Table 5** **Performance on dataset.** Krippendorff's alpha ($\alpha$) performance on datasets averaged over models and prompts, best results in bold. N total = 11,880.

| Dataset | $\alpha$(CI) n per dataset = 1980 |
|---|---|
| WorldTree v2 | **.83** (.81, .85) |
| CommonsenseQA | .71 (.68, .73) |
| OpenBookQA | .65 (.63, .68) |
| StrategyQA | .31 (.27, .36) |
| MedMCQA | .31 (.28, .34) |
| MedQA | .21 (.19, .24) |

## RESULTS

All scores within this paper are displayed with 95% confidence intervals (CI). Although the performance of many prompts averaged over all datasets is notably similar, applying reasoning strategies outperforms direct prompting. A closer examination of the results obtained from the latest model, GPT-4, highlights the advantage of employing specific prompts (Table 4). It shows the retained performance of the automatically discovered prompt by *Zhou et al. (2023b)*, which also has a notable result in the score averaged over models. Interestingly, the self-critique prompt yielded relatively low scores. It also generated multiple answers in various observed instances, which were excluded from the scoring process. The 512-token output limit was reached in only 80 of 11,800 experiments, having no significant impact on our results. This occurred primarily due to bogus sequence repetitions in smaller models or a prompt designed to mimic conversations.

Better models find WorldTree v2 and CommonsenseQA increasingly easy, while StrategyQA suffers from ambiguous items. This highlights the need to develop more refined general-knowledge datasets or employ domain-specific datasets. The two medical datasets were the most difficult to solve (Table 5).

GPT-4 and GPT-3.5-turbo performed best (Table 6). FLAN-T5 shows surprisingly good performance for its size, but its results are probably affected by data contamination: it was

**Table 6** **Performance of models.** Krippendorff's alpha ($\alpha$) performance of models averaged over datasets and prompts, best results in bold. N total = 11,880.

| Model | $\alpha$(CI) n per model = 1980 |
|---|---|
| GPT-4 | **.78** (.76, .81) |
| GPT-3.5-turbo | .62 (.59, .65) |
| Davinci-003 | .47 (.45, .50) |
| Flan-T5-XXL | .45 (.42, .47) |
| Davinci-002 | .41 (.38, .44) |
| Command-XL | .32 (.29, .35) |

**Table 7** **Performance of models per dataset.** Krippendorff's alpha ($\alpha$) performance of models per dataset averaged over prompts. Average over 330 items per model/dataset pair, best results in bold. N total = 11,880.

| Model dataset | Command-XL | Flan-T5-XXL | GPT-3.5-turbo | GPT-4 | Davinci-002 | Davinci-003 |
|---|---|---|---|---|---|---|
| CommonsenseQA | .57 (.50, .64) | .81 (.75, .85) | .70 (.64, .76) | **.82** (.76, .87) | .68 (.62, .74) | .68 (.62, .74) |
| MedQA | .06 (.01, .13) | .02 (.00, .07) | .40 (.32, .47) | **.55** (.47, .61) | .09 (.03, .15) | .17 (.11, .24) |
| MedMCQA | .08 (.01, .14) | .10 (.03, .17) | .51 (.44, .58) | **.73** (.67, .79) | .20 (.13, .27) | .21 (.14, .28) |
| OpenBookQA | .43 (.36, .50) | .69 (.63, .76) | .77 (.71, .83) | **.91** (.87, .95) | .45 (.37, .52) | .66 (.59, .72) |
| StrategyQA | .10 (.00, .21) | .23 (.12, .34) | .44 (.33, .55) | **.69** (.61, .76) | .20 (.09, .32) | .22 (.12, .31) |
| WorldTree v2 | .67 (.61, .73) | .77 (.72, .83) | .89 (.85, .93) | **.97** (.95, .99) | .84 (.79, .89) | .84 (.80, .89) |

**Table 8** **Accuracy of models per dataset.** Accuracy of models per dataset averaged over prompts. Average over 330 items per model/dataset pair, best results in bold. N total = 11,880.

| Model dataset | Command-XL | Flan-T5-XXL | GPT-3.5-turbo | GPT-4 | Davinci-002 | Davinci-003 |
|---|---|---|---|---|---|---|
| CommonsenseQA | .66 (.61, .71) | **.85** (.81, .89) | .76 (.71, .81) | **.85** (.81, .90) | .75 (.70, .79) | .75 (.70, .80) |
| MedQA | .27 (.22, .32) | .22 (.17, .26) | .53 (.47, .58) | **.65** (.60, .70) | .28 (.23, .33) | .35 (.30, .40) |
| MedMCQA | .31 (.26, .36) | .35 (.30, .40) | .63 (.58, .69) | **.80** (.76, .85) | .41 (.35, .46) | .41 (.36, .47) |
| OpenBookQA | .58 (.52, .63) | .78 (.73, .82) | .83 (.79, .88) | **.93** (.91, .96) | .59 (.54, .65) | .75 (.70, .80) |
| StrategyQA | .57 (.51, .62) | .62 (.57, .68) | .73 (.68, .79) | **.85** (.81, .89) | .63 (.57, .68) | .63 (.58, .69) |
| WorldTree v2 | .75 (.71, .80) | .83 (.79, .87) | .92 (.89, .95) | **.98** (.96, .99) | .88 (.85, .92) | .88 (.85, .92) |

instruction-fine tuned on the sub-datasets CommonsenseQA and StrategyQA. That effect shows clearly in its score on CommonsenseQA, where FLAN-T5 has a similar score to GPT-4 (Table 7). It remains an open question why the data contamination did not equally affect the score of FLAN-T5 for StrategyQA. Table 7 also shows a large performance difference on the specialized medical datasets, where only the top models GPT-4 and GPT-3.5-turbo displayed decent performance. The Krippendorff's alpha scores reveal clearly that FLAN-T5 performs merely better than chance on the dataset MedQA (GPT-4 *vs.* FLAN-T5: .55 *vs.* .02) and point out the actual performance difference to GPT-4 much better than the accuracy scores (Table 8, GPT-4 *vs.* FLAN-T5: .65 *vs.* .22).

Comparing the scores of direct prompting with all of the prompts for externalized reasoning (Table 9) shows that the models Command-XL and GPT-4 profit the most from

**Table 9** **Performance of prompts per model.** Krippendorff's alpha ($\alpha$) performance of prompts per model averaged over datasets. Average over 198 items per prompt/model pair, best results in bold. N total = 11,880.

| Model prompt | Command-XL | Flan-T5-XXL | GPT-3.5-turbo | GPT-4 | Davinci-002 | Davinci-003 |
|---|---|---|---|---|---|---|
| Direct | .26 (.18, .33) | .49 (.41, .58) | .61 (.53, .69) | .71 (.64, .79) | .41 (.31, .50) | .44 (.35, .53) |
| Kojima | .25 (.16, .34) | .46 (.38, .55) | **.66** (.57, .75) | .80 (.73, .87) | .42 (.33, .51) | .45 (.36, .54) |
| Zhou | .35 (.27, .43) | .44 (.37, .51) | .62 (.53, .71) | **.83** (.77, .90) | **.53** (.45, .62) | .50 (.41, .59) |
| Plan | .34 (.25, .42) | .45 (.37, .53) | .61 (.52, .70) | .77 (.71, .84) | .37 (.30, .45) | .46 (.37, .55) |
| Articulate | .33 (.26, .40) | **.50** (.42, .58) | .59 (.49, .68) | .79 (.71, .86) | .44 (.35, .53) | **.52** (.43, .60) |
| Rephrase | **.42** (.33, .51) | .46 (.38, .54) | .61 (.52, .70) | .78 (.71, .85) | .44 (.35, .53) | .46 (.37, .55) |
| Elaborate | .34 (.26, .42) | .42 (.33, .51) | .61 (.51, .70) | .77 (.70, .84) | .51 (.42, .60) | .43 (.35, .51) |
| Converse | .31 (.22, .40) | .44 (.35, .52) | .58 (.49, .67) | .74 (.66, .81) | .35 (.26, .43) | .46 (.38, .54) |
| Self-critique | .32 (.26, .39) | .41 (.35, .47) | .58 (.49, .68) | .76 (.69, .84) | .38 (.30, .47) | .48 (.39, .57) |
| Zhou-instruction | .38 (.30, .46) | .43 (.35, .51) | .64 (.54, .73) | .79 (.72, .86) | .33 (.26, .40) | .49 (.41, .58) |

**Table 10** **Performance of prompts per dataset.** Krippendorff's alpha ($\alpha$) performance of prompts per dataset averaged over models. Average over 198 items per prompt/dataset pair, best results in bold. N total = 11,880.

| Dataset prompt | CommonsenseQA | MedQA | MedMCQA | OpenBookQA | StrategyQA | WorldTree v2 |
|---|---|---|---|---|---|---|
| Direct | .68 (.60, .76) | .21 (.12, .30) | .28 (.18, .37) | .65 (.56, .73) | .24 (.10, .38) | .84 (.77, .90) |
| Kojima | .69 (.61, .77) | .22 (.14, .31) | .25 (.16, .35) | .61 (.52, .70) | **.46** (.32, .59) | .79 (.72, .86) |
| Zhou | .72 (.64, .79) | .23 (.14, .32) | **.37** (.27, .46) | **.74** (.66, .81) | .32 (.19, .44) | .83 (.77, .89) |
| Plan | .73 (.65, .80) | .19 (.11, .28) | .30 (.21, .40) | .65 (.56, .73) | .27 (.12, .42) | .82 (.75, .88) |
| Articulate | .72 (.64, .80) | .22 (.14, .31) | .35 (.25, .45) | .67 (.59, .75) | .27 (.13, .40) | **.88** (.83, .93) |
| Rephrase | **.75** (.68, .82) | .21 (.13, .29) | .31 (.22, .41) | .61 (.51, .70) | .42 (.30, .55) | .87 (.82, .92) |
| Elaborate | .68 (.60, .76) | **.25** (.17, .34) | .36 (.25, .45) | .64 (.56, .72) | .33 (.20, .47) | .82 (.75, .88) |
| Converse | .63 (.55, .72) | .20 (.12, .29) | .32 (.23, .41) | .63 (.55, .72) | .30 (.16, .43) | .78 (.71, .85) |
| Self-critique | .73 (.65, .80) | .19 (.11, .27) | .25 (.16, .34) | .66 (.56, .74) | .23 (.09, .37) | .82 (.75, .88) |
| Zhou-instruction | .73 (.66, .81) | .19 (.11, .28) | .26 (.17, .36) | .65 (.57, .74) | .28 (.14, .42) | .86 (.80, .92) |

externalized reasoning, whereas for FLAN-T5, direct prompting still ranks as one of its optimal methods.

The comparison of the positioning of the externalized reasoning at the end of the input "Zhou" *vs.* at the beginning of the input "Zhou-instruction" did not matter a lot for most of the models, except for Davinci-002, where putting the externalized reasoning at the end showed better performance.

Different prompts work well on specific datasets (Table 10). Comparing the two overall best prompts, "Zhou" and "Rephrase", shows that "Zhou" excels at OpenBookQA but not at StrategyQA, whereas "Rephrase" excels at StrategyQA but not at OpenBookQA. This could be because rephrasing helps untangle the ambiguous formulation of the questions in StrategyQA. The comparison of prompts on the MedQA dataset is made difficult by the low average scores achieved by several models, which makes the effects of specific prompts harder to detect.

The results reported as accuracy values can be found in the appendix.

## DISCUSSION

**Conclusion.** Our findings suggest that using reasoning strategies significantly improves performance beyond what is achieved through direct prompting alone. Interestingly, this improvement does not strongly correlate with the model's size, as both GPT-4, a very large model, and Command XL, a smaller model, show the most benefits of employing reasoning strategies. The state-of-the-art prompts developed by Zhou and Kojima demonstrate robust performance across a wide range of scenarios (*Zhou et al., 2023b*; *Kojima et al., 2022*). Comparing different prompt strategies presents challenges due to ceiling effects in larger models, which easily handle simpler datasets. In comparison, smaller models struggle with more challenging medical datasets, irrespective of the prompt strategy. As expected, GPT-4, the most powerful of the models tested, consistently outshines the others across all datasets. We found that Krippendorff's alpha is a practical and intuitively understandable metric for evaluating performance on multiple-choice datasets, proving especially useful when merging scores from datasets with varying numbers of answer choices.

**Limitations.** The presented work has several limitations. Our study aimed to test various combinations of prompts, datasets, and models under budgetary constraints. We opted to subsample datasets based on a statistical power analysis to achieve this. This limits the direct comparison of our results to evaluations on full benchmark test sets. Upon inspecting results for some academic benchmark datasets generated through crowdsourcing, we found that the quality of a sizable subset of examples was not optimal. One typical pattern we found was that questions and answer choices did not allow for clearly picking a single best answer, but multiple options were reasonable. More advanced models tend to correctly point out such problems in their reasoning response and refrain from selecting a single answer choice. We did not use methods such as self-consistency (*Wang et al., 2022*) that maximize final accuracy at the expense of practical interpretability, *i.e.,* we targeted situations in which users expect a single, high-quality and easily interpretable reasoning chain rather than a collection of noisy reasoning chains. Results achieved when using prompts in conjunction with ensemble methods might potentially differ.

Our study included state-of-the-art closed-source models, which constantly change, making replication and comparisons over time difficult. We partially address this concern by making all data generated by models at the time of our experiment openly available. The lack of documentation of closed models also leads to concerns about the contamination of training data with benchmark datasets. According to the GPT-4 technical report, the influence of data leakage during pre-training is relatively minor (*OpenAI, 2023b*). However, this assertion might not hold across the board. Fine-tuning on parts of the benchmark data would, of course, have a much bigger effect (*Zhou et al., 2023a*). Additionally, the continuous retraining of models like GPT-4 on collected usage data poses another leakage threat, as benchmark dataset examples could be reintroduced to the model in altered forms, circumventing basic string matching defenses (*Balloccu et al., 2024*). While data contamination issues do not severely impact our comparison of different prompts, we

caution against strongly interpreting results across different models. We noted that Flan-T5 (*Longpre et al., 2023*), instruction-finetuned on the subsets of CommonsenseQA and StrategyQA, outperformed GPT-3.5-turbo on CommonsenseQA.

The dataset-specific performance comparison between text-davinci-003 and text-davinci-002 presents an interesting finding. In this comparison, text-davinci-003 only demonstrates superior performance over text-davinci-002 in two of six datasets, with both models scoring equally on the other four. Remarkably, the datasets where text-davinci-003 leads are not from the same domain: it shows an advantage in the medical dataset MedQA but not in another, MedMCQA, and the scientific dataset OpenBookQA but not in WorldTree. This raises questions, as one would anticipate that the next model generation would demonstrate improved performance across datasets on the same topic.

Our objective was to evaluate the efficacy of state-of-the-art prompts across diverse models and question-answering datasets to determine their performance consistency under various conditions. To achieve this, we developed new prompts by integrating insights from recent research or adapting high-performance prompts specific to our needs. We acknowledge that prompts which are semantically similar but differ syntactically can lead to varied outcomes, as observed in medical datasets (*Liévin, Hother & Winther, 2022*). Future research could explore the comparison of semantically similar prompts, such as through paraphrasing, to further this understanding.

**Related work.** Several related studies evaluated zero-shot prompting performance. As a notable example, *Liévin, Hother & Winther (2022)* performed a comparable zero-shot CoT evaluation focused on medical datasets. Earlier work evaluating multiple models and datasets zero-shot includes commonsense data (*Zhou et al., 2020*) and assessing the performance of T0 on multiple-choice tasks (*Orlanski, 2022*). HELM (*Liang et al., 2022*) covers a wide range of model comparisons. Zero-shot reasoning can also be enhanced by generating precise reasoning steps building upon each other (*Ling et al., 2024*) or including logic dependencies for iterative verification and revision (*Zhao et al., 2023*). Our study added to current knowledge by focusing on finding simple and versatile chain-of-thought prompting approaches that work across a spectrum of models. Our included question-answering datasets go beyond simple mathematical questions that could also be solved *via* an external tool but instead focus on testing a combination of logical, verbal reasoning with real-world knowledge.

**Future work.** The current study can be extended by evaluating prompts and datasets with additional models, particularly the multitude of openly available LLMs like LLaMa, the Pythia suite, dialog-tuned models like Alpaca (*Touvron et al., 2023*; *Biderman et al., 2023*; *Taori et al., 2023*), StableLM (*Stability AI, 2023*), and OpenAssistant (*LAION, 2023*). Finally, user evaluations of the quality and explanatory utility of reasoning chains generated by different prompts and models need to be conducted.

# ACKNOWLEDGEMENTS

We thank the Cohere team for providing custom API access, enabling faster inference and unrestricted analysis of medical question datasets occasionally flagged by the standard API.

### Funding

The authors received no funding for this work.

### Competing Interests

The authors declare there are no competing interests.

### Author Contributions

- Konstantin Hebenstreit conceived and designed the experiments, performed the experiments, analyzed the data, performed the computation work, prepared figures and/or tables, authored or reviewed drafts of the article, and approved the final draft.
- Robert Praas conceived and designed the experiments, performed the experiments, analyzed the data, performed the computation work, prepared figures and/or tables, authored or reviewed drafts of the article, and approved the final draft.
- Louis P. Kiesewetter performed the computation work, authored or reviewed drafts of the article, and approved the final draft.
- Matthias Samwald conceived and designed the experiments, authored or reviewed drafts of the article, and approved the final draft.

### Data Availability

The data is available at Zenodo: Simon Ott, Konstantin Hebenstreit, Valentin Liévin, Christoffer Egeberg Hother, Milad Moradi, Maximilian Mayrhauser, Robert Praas, Ole Winther, & Matthias Samwald. (2023). ThoughtSource: A central hub for large language model reasoning data (code snapshot) (1.0) [Data set]. Zenodo. https://doi.org/10.5281/zenodo.8199390.

### Supplemental Information

Supplemental information for this article can be found online at http://dx.doi.org/10.7717/peerj-cs.1999#supplemental-information.

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
