# Peer review of "A comparison of chain-of-thought reasoning strategies across datasets and models"

_PeerJ Computer Science, doi:10.7717/peerj-cs.1999_

## Round 0.1 · original submission · Major Revisions

Both reviewers value the contribution of this work positively, however they also identify weaknesses in the experiment design which needs major revisions.

As both reviewers note, there are experimental details that are thin and/or vaguely justified. To address these, the work would need either extending the experiments through testing with more settings (e.g. more prompts) and/or a compelling justification as to why the current experiment settings have been selected and are sufficient for the purposes of the research questions set forth in this work.

Reviewer 1 also expects the findings and conclusions of the study to be made clearer, and makes additional suggestions of minor revisions.

Reviewer 1 has suggested that you cite specific references. You are welcome to add it/them if you believe they are relevant. However, you are not required to include these citations, and if you do not include them, this will not influence my decision.

**Language Note:** PeerJ staff have identified that the English language needs to be improved. When you prepare your next revision, please either (i) have a colleague who is proficient in English and familiar with the subject matter review your manuscript, or (ii) contact a professional editing service to review your manuscript. PeerJ can provide language editing services - you can contact us at [email protected] for pricing (be sure to provide your manuscript number and title). – PeerJ Staff

·

Basic reporting

1. Writing
Usually, the quote symbol should be ``'' for latex instead of "". Please fix all the appearance across the paper.

2. Presentation
It would be better to highlight the best scores in tables to make the message clearer.

3. Literature
In the Method section, the authors mentioned the implemented zero-shot methods and also provided the corresponding prompts, but it would be better to provide more information about the difference, e.g. where the prompt will be located, maybe a figure to illustrate.
Though not all of the zero-shot CoT can be implemented, including more, beyond Table 3, in related work will be beneficial, e.g. [1] [2].

[1] Zhao, Xufeng, Mengdi Li, Wenhao Lu, Cornelius Weber, Jae Hee Lee, Kun Chu, and Stefan Wermter. "Enhancing Zero-Shot Chain-of-Thought Reasoning in Large Language Models through Logic." arXiv preprint arXiv:2309.13339 (2023).

[2] Ling, Zhan, Yunhao Fang, Xuanlin Li, Zhiao Huang, Mingu Lee, Roland Memisevic, and Hao Su. "Deductive Verification of Chain-of-Thought Reasoning." arXiv preprint arXiv:2306.03872 (2023).

Experimental design

1. Experimental setting
The max_token is set universally to 512, which may be not enough for hard question that requires complex reasoning procedures; and it is also not friendly for CoT methods that require revision. I am curious about whether there is any deduction error caused by this token limitation instead of the prompting method itself. I also suggest prolonging the context length for a more diverse dataset.

2. Dataset selection
The selection of the dataset can be more diverse, e.g. including math reasoning, symbol reasoning, etc.

Validity of the findings

This work aims to empirically investigate zero-shot CoT prompting performance across models and datasets within a limited budget. It is a solid work with many experiments verified.

Further suggestions would be
1) a clearer conclusion for the readers. Now there are only scattered messages, but an extraction of some common information, e.g. whether CoT revision is necessary or whether self-check is generally helpful etc, would be helpful.
2) a discussion of the possibility of the data leaking may be also helpful. It is commendable that the authors highlighted in the Discussion section the fine-tuning of Flan-T5 with certain datasets, a critical aspect for ensuring fairness in method comparisons. Indeed, GPT-3.5 or GPT-4 may also have data leaking issues.

Additional comments

Readers would benefit more if the authors could make the message clearer and bring more informative insights, for example, take-home messages when designing prompts.

Reviewer 2 ·

Basic reporting

This paper empirically compares different chain-of-thought strategies on different datasets. The experiments are comprehensive in terms of datasets and methods covered, but the conclusions are hard to generalize to unseen datasets.

Experimental design

The experiments of different datasets are comprehensive, however,
- The power analysis part is confusing and not necessary, in the context of a natural language processing paper.
- It's hard to conclude whether these prompts are optimal or not. For example, would a paraphrase of any of the prompt work better? To this end, the conclusion of this paper is hard to be generalized to more datasets.

Validity of the findings

As mentioned above, the findings are obviously valid, but it's unclear how generalizable the findings are to a broader set of tasks. I would suggest the authors experiment with prompts that are paraphrase of the current ones, and investigate whether different forms of the same meaning matter for the performance.

Additional comments

A minor suggestion on table presentation: it would be good to boldface the best number in a column/row, whichever appropriate, to present the result more clearly.

Cite this review as

---

## Round 0.2 · Minor Revisions

One of the original reviewers is positive about the changes made in the manuscript, but also suggests some minor revisions to be considered prior to publication. I encourage the authors to consider these revisions to the extent possible and to provide a response to the reviewer suggestions.

·

Basic reporting

After the revision, the submission now fixed many misuses of symbols, e.g. quoting with ``'' instead of "", and typos. But there are still many single-quoted texts, e.g. 'cot trigger' in the last paragraph of section Methods/Datasets, which looks weird (in terms of both rendering with '' and meaning/reason of using a single quote) and deserves a second consideration.

Experimental design

1. It would be valuable if the detailed budget of the experiments could be reported, including openai API usage and local model run time, device, etc.

2. Potentially additional experiments would be a larger token limit with complicated reasoning strategies, e.g. ensemble-based ones or iterative prompting methods, for those cases, 512 is far less than enough.

Validity of the findings

This work systematically tests various Chain-of-Thought prompting strategies, across many datasets. The applied Krippendorff’s method is also helpful, especially for massive evaluation of models and tasks.

---

## Round 0.3 · accepted · Accept

I appreciate the authors for addressing the minor revisions suggested in the previous round, having modified the quotation marks and added details of experiment costs. I recommend that the paper can now be accepted in its present form.